# The Role of Verbal Feedback in the Motor Learning of Gymnastic Skills: A Systematic Review

**Marcin Starzak** [1] , **Michał Biegajło** [1] , **Marta Nogal** [1] , **Tomasz Niźnikowski** [1] , **Tadeusz Ambroży** [2] , **Łukasz Rydzik** [2,*] **and Jarosław Jaszczur-Nowicki** [3,*]

1   Biała Podlaska Faculty of Physical Education and Health, Józef Piłsudski University of Physical Education, 00-968 Warszawa, Poland; marcin.starzak@awf.edu.pl (M.S.); michal.biegajlo@awf.edu.pl (M.B.); marta.nogal@awf.edu.pl (M.N.); tomasz.niznikowski@awf.edu.pl (T.N.)
2   Institute of Sports Sciences, University of Physical Education, 31-571 Krakow, Poland; tadek@ambrozy.pl
3   Department of Tourism, Recreation and Ecology, University of Warmia and Mazury in Olsztyn, 10-719 Olsztyn, Poland
*   Correspondence: lukasz.rydzik@awf.krakow.pl (Ł.R.); j.jaszczur-nowicki@uwm.edu.pl (J.J.-N.)

**Abstract:** The main purpose of this study was to systematically review the effects of feedback on motor skill acquisition in gymnastic skills. A systematic literature search was conducted in the electronic databases MEDLINE (EBSCO), Scopus, SPORTDiscus (EBSCO), and Web of Science. Of the initial 743 search articles, 13 studies were included in the quantitative analysis. Studies were included if they met the following criteria: (a) healthy subjects, (b) studies with gymnastic, artistic gymnastic, or trampoline elements in the study protocol, (c) verbal feedback intervention, (d) the study compared verbal feedback intervention with other forms of feedback, instructional intervention, or with a control intervention, and (e) task performance evaluation. Methodological quality was identified using the PEDro scale. Combining verbal instructions with different forms of feedback is beneficial when learning complex gymnastic skills. Verbal feedback may be useful to improve the technical performance of a gymnastic element; in particular, information regarding the errors committed in a key element of the performance seems to be effective in the motor learning process.

**Keywords:** motor learning; feedback; gymnastic element; verbal instruction

## 1. Introduction

The sport of gymnastics is changing very dynamically. The difficulty of single exercises, their combinations, and whole routines is increasing [1,2]. However, a successful gymnastics routine is more than a series of well-executed individual movements. Transitions between individual elements must be executed smoothly so that the entire performance "flows" as a coherent unit. Most serial activities of reasonably long duration are characterized by an inherent timing or rhythmic structure among certain components. The coach must be careful to identify the components within the routine that go together and have the athletes practice them as a unit so as not to disrupt essential timing [3–5].

Therefore, the methods for improving the learning of motor skills are sought by many scholars and practitioners [6]. In the domain of sports, coaches have looked for ways of being more efficient in their teaching and maximizing their results. One method that has appeared to be effective theoretically is the use of verbal feedback to support the acquisition of motor skills in gymnastics.

Very often it is argued that visual presentation is preferred over verbal instruction because vision is more proficient than language at specifying precise aspects of human movement [7]. In addition, vision may enhance a performer's ability to detect and correct errors [8]. However, it is important to note that coaches in gymnastics use verbal feedback during both training and competition. This is because, on the one hand, coaches are not able to perform difficult elements or routines, but on the other hand, they can use verbal

feedback quickly and easily, and it can be provided to the learner simultaneously with performance.

Currently, providing verbal feedback to aid learning is used extensively in gymnastics. Studies exist on the efficacy of verbal feedback not only in gymnastics [9], skiing [10], dance [11], swimming [12], basketball [13], and golf [14], but also in other areas. However, there are vastly discrepant findings regarding the efficacy of verbal feedback as a learning tool. Early research attempted to quantify the effectiveness of verbal feedback in motor skill acquisition [15–17].

According to Salmoni et al. [18], it is necessary to look for effective methods of providing feedback in order to improve the process of teaching and learning motor skills. It should be noted that feedback may be provided in a variety of ways, and the effectiveness of teaching and learning motor skills depends on the manner we choose [19–24]. For instance, providing verbal feedback on errors and ways of correcting them is an effective approach, particularly when teaching motor skills to beginners [25]. In turn, Niźnikowski and Sadowski [26] claim that when it comes to teaching complex motor skills to experienced gymnasts, it is necessary to identify key information and provide it to them during the skill acquisition process. Sadowski et al. [27] found that bandwidth feedback proved to be more effective than feedback on all errors.

So far, the best way to make use of different types of feedback in teaching and learning motor skills of various complexities and degrees of difficulty is not established [22–24]. Schmidt and Lee [28] claim that further research is needed to establish relations between the level of difficulty of a given motor skill and the type of feedback. There is still a scarcity of empirical data on the effects of different types of feedback on the effectiveness of teaching and learning motor skills of various complexities and degrees of difficulty [10,13,29–32]. There is also not enough research on effective descriptive or prescriptive verbal feedback in learning motor skills in gymnastics. In the case of a descriptive statement, only the error a person made during the performance of a skill is described. However, a prescriptive statement describes errors made during the performance of a skill and states (i.e., prescribes) what needs to be done to correct them. Therefore, the question is whether to use descriptive verbal knowledge of performance or prescriptive verbal knowledge of performance?

The purpose of this study was to systematically review the effects of feedback on motor skill acquisition in gymnastic skills. Specifically, the study sought to summarize the evidence of (1) the effectiveness of feedback and (2) the effects of feedback elements (i.e., prescriptive knowledge of performance and descriptive verbal feedback knowledge of performance) on the skill acquisition of gymnasts in the course of the training process. The findings of this review can provide simple and clear guidance on feedback for coaches, which has been shown to be highly effective for promoting athletes' motor skill learning.

## 2. Materials and Methods

### 2.1. Search Strategy

The preferred reporting items for systematic reviews and meta-analysis statement (PRISMA) guidelines for reporting a systematic review were adopted [33]. Three investigators (M.S., M.N., and M.B.) searched the following electronic databases without time restriction. A literature search was conducted on 14 January 2022. The four databases: MEDLINE (EBSCO), Scopus, SPORTDiscus (EBSCO), and Web of Science were utilized for the search process. The search was limited to peer-reviewed articles that were published in the English language. The following keywords with Boolean operators were applied: ("verbal feedback" or "verbal instruction*" or "verbal information" or "verbal augment* feedback" OR "verbal guidance") and (trampoline* or acrobat* or gymnast*).

### 2.2. Eligibility Criteria

Primary research articles were selected if they met the following inclusion criteria: (a) healthy subjects, (b) studies with gymnastic, artistic gymnastic, or trampoline elements in the study protocol, (c) verbal feedback intervention was used, (d) the study compared

verbal feedback intervention with other feedback, instructional intervention, or with a control intervention (e.g., not use any feedback), and (e) task performance evaluation (e.g., quality of movement) was tested. Exclusion criteria were: (a) no randomized study design, (b) not motor learning protocol, and (c) no comparison group.

### 2.3. Study Selection

After complementation of the articles, the duplicates were removed automatically in EndNote. Other duplicates were deleted manually. The selection process was performed independently by two reviewers (M.N. and M.B.). In any case of disagreement, ambiguous issues were discussed with a third independent reviewer (M.S.).

### 2.4. Data Extraction

Three reviewers (T.N., M.N., and M.B.) independently extracted data regarding design, participants (e.g., sex, age, characteristics), intervention (e.g., description of verbal feedback intervention), motor learning intervention protocol (e.g., duration of practice, volume, and retention test), outcome measures, and main results. A Cohen's kappa score was calculated to determine the level of agreement between reviewers.

### 2.5. Quality Assessment

The quality of included studies was assessed using the Physiotherapy Evidence Database (PEDro) quality scale [34]. The PEDro scale consists of 11 criteria that assess the methodological quality of experimental studies. Each criterion can be rated from 0 to 1 point. For this review, criterion 1 was not included as it comprises the external validity, therefore a maximum possible score of 10 points could be reached. Points were awarded only when a given criterion was clearly satisfied (Table 1). All included studies were independently assessed by two raters (M.S., M.N.). The agreement between the reviewers was assessed with Cohen's kappa. In case of any ambiguous issues regarding rating points, a final consensus was reached by discussion between the reviewers. Scores within the range of 8–10 were regarded as "excellent", 6–8 as "good", 4–5 as "fair", and ≤3 as "poor" quality.

**Table 1.** PEDro quality rating scores *.

| Study | Criterion | | | | | | | | | | | PEDro Score |
|---|---|---|---|---|---|---|---|---|---|---|---|---|
| | 1 | 2 | 3 | 4 | 5 | 6 | 7 | 8 | 9 | 10 | 11 | |
| Amri-Dardari [35] | 1 | 0 | 0 | 0 | 0 | 0 | 0 | 0 | 0 | 1 | 1 | 2 |
| Barić and Busko [36] | 0 | 1 | 0 | 0 | 0 | 0 | 0 | 1 | 0 | 1 | 1 | 4 |
| Frikha et al. [37] | 0 | 1 | 0 | 0 | 0 | 0 | 1 | 0 | 0 | 1 | 1 | 4 |
| Maleki et al. [38] | 0 | 1 | 0 | 1 | 0 | 0 | 0 | 0 | 0 | 1 | 1 | 4 |
| Niźnikowski and Sadowski [26] | 0 | 1 | 0 | 1 | 0 | 0 | 0 | 0 | 0 | 1 | 1 | 4 |
| Niźnikowski et al. [39] | 0 | 1 | 0 | 1 | 0 | 0 | 0 | 0 | 0 | 1 | 1 | 4 |
| Niźnikowski et al. [40] | 0 | 1 | 0 | 1 | 0 | 0 | 0 | 0 | 0 | 1 | 1 | 4 |
| Niźnikowski and Nogal [9] | 0 | 1 | 0 | 0 | 0 | 0 | 0 | 0 | 0 | 1 | 0 | 2 |
| Nogal and Niźnikowski [41] | 0 | 1 | 0 | 1 | 0 | 0 | 0 | 0 | 0 | 1 | 0 | 3 |
| Potdevin et al. [42] | 0 | 0 | 0 | 1 | 0 | 0 | 0 | 0 | 0 | 1 | 1 | 3 |
| Sadowski et al. [27] | 0 | 1 | 0 | 0 | 0 | 0 | 0 | 0 | 0 | 1 | 1 | 3 |
| Sadowski et al. [43] | 0 | 1 | 0 | 1 | 0 | 0 | 0 | 0 | 0 | 1 | 1 | 4 |
| Wali-Menzli et al. [44] | 0 | 1 | 0 | 0 | 0 | 0 | 0 | 0 | 0 | 1 | 1 | 3 |

* PEDro rating criteria: (1) eligibility criteria were specified, (2) subjects were randomly allocated to groups, (3) allocation was concealed, (4) the groups were similar at baseline regarding the most important prognostic indicators, (5) there was blinding of all subjects, (6) there was blinding of all therapists who administered the therapy, (7) there was blinding of all assessors who measured at least one key outcome, (8) measures of at least one key outcome were obtained from more than 85% of the subjects initially allocated to groups, (9) all subjects for whom outcome measures were available received the treatment or control condition as allocated, (10) the results of between-group statistical comparisons are reported for at least one key outcome, (11) the study provides both point measures and measures of variability for at least one key outcome.

## 3. Results

### 3.1. Study Selection

Figure 1 shows the flow chart of the study selection process. The initial search yielded a total of 743 records. Screening for titles and abstracts resulted in the identification of 55 possibly relevant papers. Eventually, after the full-text screening, 13 studies were included in this systematic review. Consistency between reviewers was 0.91 for the data extraction review process (percentage agreement = 91%).

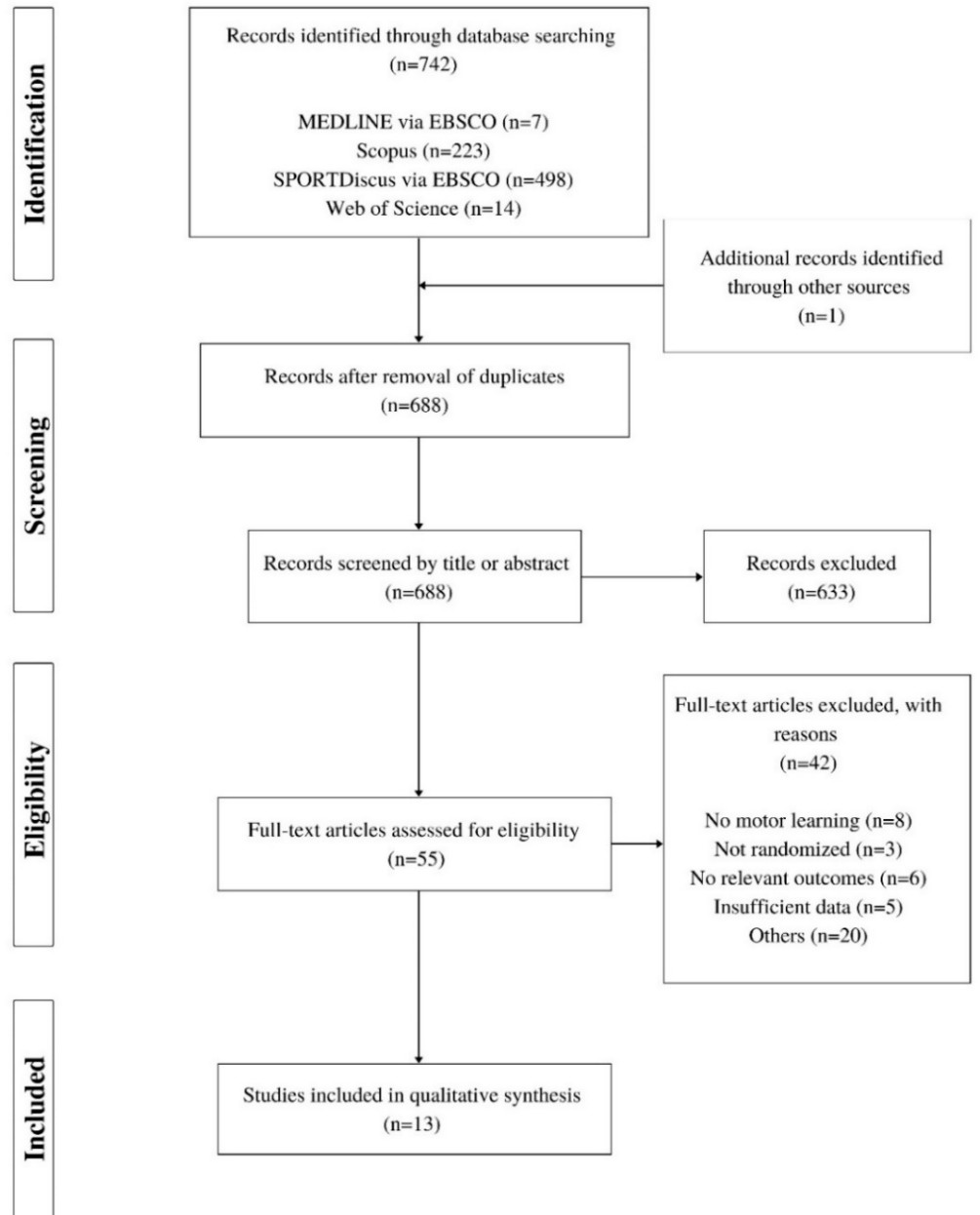

**Figure 1.** Flow chart of the study selection process.

### 3.2. Study Characteristics

For each included study, the characteristics of data extraction are shown in Table 2.

**Table 2.** Summary of included studies.

| Study | Population | Sex (n), Mean Age (Years ± SD) | Feedback Intervention | Motor Learning Protocol | | | Outcome Measures | Main Results |
|---|---|---|---|---|---|---|---|---|
| | | | | Gymnastic Skill | Volume | Retention Phase | | |
| **Amri-Dardari et al.** [35] | Physical education students | M (n = 135), 20.45 ± 1.1 | Group 1: verbal feedback with technical instructions, safety, explanatory drawings/sketches, and partial demonstrations of the teacher (TG)<br>Group 2: self-modeling, expert modeling, and model's superposition at each session, in addition to a classical learning based on verbal feedback (MG)<br>Group 3: self-modeling and mathematical simulation/virtualization of their movement, in addition to a classical learning based on verbal feedback (SG) | Vaulting jump on the vault table | 12 weeks, 2 sessions/week, 1 h 30 min per session | No | TP: 20-point scale | Pre-Post:<br>TG, MG, and SG ↑ TP<br>MG > TG<br>MG > SG<br>SG > TG |
| **Barić and Busko** [36] | Kinesiology students | F (n = 33), 19.4 ± 0.8 | Group 1: videotaped expert model performing the skill (VD)<br>Group 2: recorded verbal instructions that described a skill (VI)<br>Group 3: demonstration and verbal instructions (C) | Rope jumping (rhythmic gymnastics) | 6 × 8 trials | No | TP: 5-point scale | Pre-Post: VD, VI, and C ↑ TP,<br>AQ: VD and C > VI |
| **Frikha et al.** [37] | Physical education students | M (n = 48), 25.0 ± 3.3 | Group 1: verbal augmented feedback (VAF)<br>Group 2: haptic augmented feedback (HAF)<br>Group 3: verbal and haptic feedback (CAF)<br>Group 4: an explanation and demonstration of the considered gymnastic elements at the beginning of the teaching process (CON) | Parallel bars task | 2 × 90 min (20 reps per element) | 2 days | TP: 10-point scale<br>Self-perceived task difficulty (PD): 15-point scale<br>Self-perceived competence (PC): 7-point scale | AQ: CAF, HAF, and VAF ↑ TP<br>CAF > VAF<br>RE: CAF > VAF and HAF, CAF ↓ PD to VAF, and HAF<br>AQ: ↑ PD CON to VAF, and HAF<br>RE: CAF ↓ PD to VAF and HAF<br>CAF ↑ SC to VAF, and HAF<br>RE: CAF ↑ SC to VAF, HAF, and CON |
| **Maleki et al.** [38] | Amateur gymnastic students of Physical Education | M (n = 50), 20.35 ± 1.44 | Group 1: observed the execution of real model without any interference (AOG)<br>Group 2: observed the execution of real model with verbal descriptions by coach (AOVG)<br>Group 3: observed the demonstration of animated model combined with verbal descriptions by coach (AONG) | Handstand | 3 weeks, 3×/week, 10 trials each session | 48 h | TP: 10-point scale | Pre-Post:<br>AOVG, AONG, and AOG ↑ TP<br>AQ:<br>AOVG and AONG > AOG<br>RE:<br>No sig between groups |
| **Niźnikowski and Sadowski** [26] | Skilled and highly skilled gymnasts | F (n = 16), 20 ± 2.35 | Group 1: received immediate verbal information about faults committed in the key elements of the mastered motor task (EG)<br>Group 2: received information about all committed errors in each attempt and how to correct them—100% feedback (CG) | Round-off–double salto backward tucked during beam dismount | 6 weeks, 3 sessions/week (90 min/session: 3 sets of 5 reps) | 6 days | TP: 10-point scale | Pre-Post: EG, and CG ↑ TP<br>AQ: EG > CG<br>RE: EG > CG |

**Table 2.** *Cont.*

| Study | Population | Sex (n), Mean Age (Years ± SD) | Feedback Intervention | Motor Learning Protocol | | | Outcome Measures | Main Results |
|---|---|---|---|---|---|---|---|---|
| | | | | Gymnastic Skill | Volume | Retention Phase | | |
| **Niżnikowski et al.** [39] | Physical education students | EG: M (n = 7), 20.4 ± 1.2 PG: M (n = 6), 20.3 ± 1.3, E&P: M (n = 7), 20.3 ± 1.1 | Group 1: verbal feedback only on errors (EG) Group 2: verbal feedback only on correct movement execution (PG) Group 3: verbal feedback on errors and how to improve them (E&G) | Vertical jump task | 6 weeks, 3×/week/60 min each session (20 sets of 5 reps | 7 days | TP: 10-point scale | Pre-Post: EG > PG, EG > E&P, PG > E&P RE: EG > PG, EG > E&P, PG > E&P |
| **Niżnikowski et al.** [40] | Young acrobatic gymnasts | EG: F (n = 9), 7.3 ± 1.3 PG: F (n = 10), 7.5 ± 1.2 E-P: F (n = 10), 7.5 ± 0.5 | Group 1: verbal feedback on performance errors (EG) Group 2: verbal feedback of performance correctness (PG) Group 3: verbal feedback on performance errors and correctness (E-P) | Backward roll | 4 weeks: 3×/week/90 min—3 sets of 5 reps | 1 week | TP: 10-point scale | Pre-Post: PG > E-P PG > EG E-P > EG RE: EG > E-P EG > PG E-P > PG |
| **Niżnikowskiand Nogal** [9] | Skilled and highly skilled gymnasts | F (n = 16), 20 ± 2.35 | Group 1: urgent verbal information about the errors committed in the key elements of the mastered motor action (EG) Group 2: instructions about all the faults (CG) | Double salto backward piked to dismount from uneven bars | 240 attempts | 6 days | TP: 10-point scale | Pre-Post: EG, and CG ↑ TP AQ: EG > CG RE: EG > CG |
| **Nogal and Niżnikowski** [41] | Young acrobatic gymnasts | F (n = 45), 7.5 ± 1.3 | Group 1: verbal feedback on errors concerning the whole motor skill performance (EWS) Group 2: verbal feedback on errors made in particular phases of the skill performance (EPS) Group 3: verbal feedback on errors that occurred in key elements (EKE) | Pike jump–trampoline | 6 weeks, 3 sessions/week (90 min/session: 3 sets of 5 reps) | 6 days | TP: 10-point scale | Pre-Post: EWS, EPS, and EKE ↑ TP AQ: EWS < EKE, EPS < EKE RE: EPS < EKE |
| **Potdevin et al.** [42] | Secondary school students | VFB: M (n = 8), F (n = 10), 12.4 ± 0.5 CON: M (n = 13), F (n = 12), 12.6 ± 0.4 | Group 1: visual (computer) feedback on his/her performance, and verbal discussion with technical advise (VFB) Group 2: verbal cues only (CON) | Handstand to flat back landing | 5 weeks/5 lessons × 120 min/15 reps each session | No | Kinematical analysis of arm-trunk angle [deg.] after 5, 10 and 15 reps, mean value of each 5 lessons (KA) Self-assessment ability (SA): correct answers of 5 question on each lesson-5-point scale Motivation evaluation (ME): Situational Motivation Scale questionnaire (intrinsic, extrinsic and motivation) | KA: VFB > CON SA: VFB > CON ME: VFB > CON |

**Table 2.** *Cont.*

| Study | Population | Sex (n), Mean Age (Years ± SD) | Feedback Intervention | Motor Learning Protocol | | | Outcome Measures | Main Results |
|---|---|---|---|---|---|---|---|---|
| | | | | Gymnastic Skill | Volume | Retention Phase | | |
| **Sadowski et al.** [27] | Athletes | M (n = 30), 11.0 ± 0.3 | Group 1: prescriptive verbal feedback about errors and how to correct them only in errors in key elements of task (B) <br> Group 2: prescriptive verbal feedback about errors and how to correct all of them in the task-100% feedback (C) | Back tuck salto after round-off | 16 weeks, 4 session/week, 10 trials each session | 1 day (RE) 1 week (DRT) | TP: 10-point scale | TP: <br> AQ: B > C <br> RE: B > C <br> DRE: B > C |
| **Sadowski et al.** [43] | Physical education students | E&P: M (n = 7), 20.3 ± 1.1, PG: M (n = 6), 20.4 ± 1.2 | Group 1: verbal information on errors and on how to correct them (E&P) <br> Group 2: verbal feedback on the correctness of performance only (PG) | Vertical jump task | 18 workouts: 60 min (20 reps/task) | 1 day | TP: 10-point scale | Pre-Post: PG ↑TP <br> RE: PG > E&P |
| **Wali-Menzli et al.** [44] | Exercise Science and Physical Education | F (n = 42), 20.6 ± 1.3 | Group 1: visualization of the perfect performance of the task (EVI) <br> Group 2: verbal instructions (VFM) <br> Group 3: video recorded exercise of the performance (VM) | The roll backward to handstand | 6 training sessions | No | TP: 10-point scale | Pre-Post: <br> EVI ↑ TP <br> VFM ↑ TP <br> VM ↑ TP <br> No sig between group in TP |

M—male, F—female, TP—task performance evaluation, AQ—acquisition phase, RE—retention test, DRE—delayed retention test, ↑ increased, ↓ decreased.

The total sample size was 520 subjects (316 men, 61%; 204 women, 39%). In eight studies [35–39,42–44] the participants were secondary school or university students of kinesiology or physical education. The subjects were amateurs with little to no prior experience in gymnastics. However, most studies state that subjects were healthy and physically active, therefore they could safely participate in the training protocols. Five other studies [9,26,27,40,41] concerned skilled athletes with different experience levels in gymnastics. In three experiments [27,40,41] the participants were young athletes (mean age range from 7.5–11.0 years old). One study [27] reported experience ranging from 1 to 2 years. The other two studies [40,41] do not explicitly describe subjects' experiences, although in Niźnikowski et al. [40] authors mentioned that participants were at the early stage of training.

Training program durations varied from 1 day to 16 weeks, with a range of 40 to 640 repetitions per element. In the majority of studies (9 of 13), the retention test for the persistence of the learning process was used. One study [27] evaluated a learning effect in retention and delayed retention tests. Four papers did not use the retention phase and provided only the pre- and post-test comparison.

Except for the one study [42], results were presented with a quantitative assessment of the technical execution of the gymnastic task. Most studies utilized the criteria of assessment based on the International Gymnastics Federation (FIG) rules. In most cases, they used an expert or judge's performance evaluation.

### 3.3. Methodological Quality

The quality analysis found that most studies were of fair methodological quality with ratings ranging from 2 to 4 (mean 3.4 ± 0.8). The mean kappa agreement between pairs of reviewers was 0.92 (percentage agreement = 92%). Across the 13 studies, the most common missing criteria were the lack of blinding methods (subjects, therapists, and assessors), concealed allocation, completeness of follow-up, and intention-to-treat analysis. Only in one study [37] was it specified that the assessors of outcomes were blinded to group allocation. Except for one study [36], all studies fulfill the criteria of obtained outcome data for at least 85% of subjects initially allocated to groups. All of the studies reported between-group statistical comparisons for at least one key outcome.

### 3.4. The Effects of Verbal Feedback in Learning Gymnastic Tasks

Of the 13 studies, six [35–38,42,44] investigated the effects of verbal instruction, non-verbal feedback, or combined on the motor learning of gymnastic skills.

Frikha et al. [37] studied whether the verbal augmented feedback, haptic augmented feedback, combined, or no instruction would be greater for improving the learning of the gymnastic task performed on the parallel bars. The subjects with the combination of the verbal and haptic feedback showed significantly higher evaluation performance scores than those with instructions provided alone, both for the acquisition and retention phase.

Barić and Busko [36] conducted a study in which they compared verbal instruction with visual presentation of the expert model and the combined conditions. In all groups, they found significantly higher values of performance grades after the eight blocks of the learning process. However, the participants in the visual and demonstration groups reached significantly better scores than the verbal group, meaning that the efficiency of learning between the intervention was different.

These results are partly in line with a study conducted by Wali-Menzli et al. [44] where the objective was to determine the most effective method for improving the performance of the roll backward to handstand skill. They compared a visual external imagery modality with visualization of their performance to comments and verbal feedback only. Although each group improved the technical execution of the task, there was no significant interaction between groups.

The effectiveness of the different feedback strategies in learning a motor skill for the performance of the vaulting jump was studied by Amri-Dardari et al. [35]. They compared

a traditionally verbally received instruction with learning by combining verbal cues with an intervention that followed self-modeling and self-modeling with a simulation in addition to verbal feedback for both groups, respectively. The results show a significant enhancement in the technical performance of each group, while the self-modeling group was better than the others.

In the study of Potdevin et al. [42], the participants in the experimental group received verbal instructions in addition to visual feedback on their performance, while in the control group subjects were verbally advised only. They found that combining verbal and visual feedback resulted in increased reported outcomes in self-assessment ability and motivation evaluation. A significant change was also seen in the kinematic assessment of the arm–trunk angle values indicating a crucial role of visual aid feedback in the traditional use of verbal instructions in optimizing motor learning.

The objective of one study [38] was to compare the effects of the different visual observations with verbal feedback. Their study aimed to assess the influence of the visual observation of the real model with or without additional verbal instructions and an animated model with verbal descriptions on motor learning of a gymnastic handstand. The results showed that all three groups improved their performance in the handstand skill from pre-test to the acquisition and retention phases. However, there was no significant difference between groups.

### 3.5. Different Forms of Verbal Feedback in Learning Gymnastic Tasks

The seven studies [9,26,27,39–41,43] investigated the effects of different types of verbal feedback on motor learning in gymnastics. Niźnikowski et al. [39,40] studied whether the use of verbal feedback based on errors committed during a performance was more efficient than verbal feedback focused on performance correctness. In both studies, the information about the faults made during the performance resulted in better technical scoring grades in the acquisition and retention phase than feedback on the correct execution of the task or combined conditions.

The efficiency of using different verbal information for learning a vertical jump task was studied by Sadowski et al. [43]. They compared the verbal information about faults and how to correct them with verbal feedback on the correctness of performance only. They observed statistically significant performance improvement for both groups in the post-test and the retention test, respectively. The participants who received feedback on the correctness of performing the task achieved better results than the subjects from the group who received information on errors and ways of correcting them. Neither group showed a score improvement in the retention phase.

The other two studies [26,27] examined the effects of different frequencies of verbal feedback in the motor learning process of skilled gymnasts. They compered a verbal instruction regarding errors made in the key elements of the task with information about all the errors made in the process of mastering the gymnastic action. In Sadowski et al. [27], the participants who received instructions focused on key elements of the task significantly enhanced the performance of their back tuck salto after roundoff in comparison to those who received all task information. In line with the study of Sadowski et al. [27], Niźnikowski and Sadowski [26] examined similar interventions in the learning of roundoff double salto backward tucked during beam dismount. They revealed that verbal feedback on errors in key elements was more efficient in mastering a motor skill compared to receiving information about all faults during the performance.

Similarly, in a study conducted by Niźnikowski and Nogal [9], they compared groups learning how to perform a double salto backward piked dismount from uneven bars. One group received verbal feedback about the errors made in the key elements of the task, while the other group received instructions about all faults made during the task. The participants significantly increased their scoring points after the course of learning in both groups, respectively. A higher-scoring evaluation was reported when participants received

feedback on errors only in the key elements compared to information on faults committed during the performance.

Finally, Nogal and Niźnikowski [41] compared the verbal feedback on errors made during the whole performance of the skill, the particular phases, and key elements. Young female acrobatic gymnasts showed significant improvement in each of the conditions, however, the information on the errors that occurred during key elements were found more efficient in mastering motor performance in the pike jump on a trampoline.

## 4. Discussion

The purpose of this study was to systematically review the effects of feedback on motor skill acquisition in gymnastic skills. Specifically, the study sought to summarise the evidence of (1) the effectiveness of feedback and (2) the effects of feedback elements (i.e., prescriptive KP and descriptive verbal feedback KP) on skill acquisition in gymnasts in the course of the training process. The findings of this review can provide simple and clear guidance on feedback for coaches, which were shown to be effective for promoting athletes' motor skill learning.

### 4.1. The Effects of Verbal Feedback in Learning Gymnastic Tasks

In our review, six studies investigated the effects of verbal instruction, non-verbal feedback, and a combination of the two on the motor learning of gymnastic skills.

Analyzing the works in this review, it can be concluded that verbal feedback reinforced by another type of feedback has a positive effect on the process of learning gymnastic skills [36,37,42]. Hebert and Landin [15] found that giving feedback in the form of verbal cues facilitates the performance of a task. Others add that this is because the learner is able to notice aspects of the movement that may be omitted when receiving visual information.

However, the effectiveness of the motor learning process of gymnastic tasks is influenced by many factors, including complexity of motor skills, the learner's level of advancement, as well as internal and external factors. We should agree with the authors Williams and Hodges [45], who showed that the effectiveness of learning may depend on how feedback is provided, and Wulf and Shea [46], who noticed that the principles used in teaching less complex tasks cannot be transferred directly to complex tasks. Laguna [47] shows that the selection of the most effective types of feedback is specific to each task. The variety of factors influencing the effectiveness of the motor learning process makes it difficult to summarize the many conclusions drawn from previous scientific studies [23,24,46–48].

### 4.2. Different Forms of Verbal Feedback in Learning Gymnastic Tasks

In this review, the seven studies investigated the effect of different types of verbal feedback on motor learning in gymnastics.

In three studies [39,40,43], attempts were made to determine which type of verbal feedback is more effective: prescriptive or descriptive. Niźnikowski et al. [39] and Niźnikowski et al. [40] studied whether the use of verbal feedback based on errors committed during performance was more efficient than verbal feedback focused on performance correctness. In turn, Sadowski et al. [43] showed that the participants who received feedback on the correctness of the performed task achieved better results than the subjects from the group who received information on errors and ways of correcting them.

Kernodle and Carlton [25] claim that providing the learner with verbal information and how to correct it is effective in the early stages of learning. Smith and Davies [49] showed that feedback on motor skill errors can have a positive impact on the self-confidence of young athletes. On the other hand, Lee et al. [50] argue that positive verbal cues with feedback instructions are beneficial in non-specific tasks.

In the next four works, attempts were made to show whether it is better to provide learners with a large amount of information or to reduce verbal feedback to a smaller

amount. There is an advantage in using feedback on errors in key elements of motor activity [9,26,27,41].

Based on the review, it was found that verbal feedback regarding errors in key elements of sports techniques in gymnastic skills is most effective in the post-test and retention-test. This may indicate that providing feedback on the most important and characteristic postures and positions of the body can bring more effective results. Learners can then focus their attention on correcting specific errors that were brought to their attention in the feedback. On the other hand, feedback that is too general and too extensive may disrupt the learning process. The learner will not be able to use redundant feedback, especially in the early stages of learning [51,52]. It was also found that the use of feedback on the key elements of sport techniques in learning complex motor skills determines the quality of their performance [53]. This is in line with the present research results. Wulf et al. [54] stated that when learning complex motor skills, a relatively small amount feedback is recommended, especially in the early stages of learning. Tzetzis et al. [55] found that less feedback translates into the sustainability of learning outcomes. This is confirmed by the results of this research.

### 4.3. Limitations and Directions for Future Research

Some limitations inherent within the review should be noted. This review was limited by the small number of available research studies. Despite the thorough literature search, a few published studies were possibly overlooked because of the keywords that may differ from those used in the current work. In addition, in some cases missing data made a systematic evaluation difficult. Moreover, a small number of studies limited the ability to draw definite conclusions from this review.

The quality of included studies was overall fair to poor, indicating low internal validity. In the majority of the studies, basic methodological procedures were inadequate or lacked crucial data. For example, incomplete reporting of study designs and interventions increases the risk of publication bias and leads to difficulties in the interpretation and generalization of the results.

Further research is needed to determine the principles of learning complex gymnastics routines. Therefore, it is advisable to carry out studies into complex gymnastic elements and routines.

### 5. Conclusions

In summary, the findings of this systematic review indicate that combining verbal instruction with other forms of feedback could be beneficial for learning complex gymnastic skills. Results also support the idea that verbal feedback may be useful to improve the technical performance of a gymnastic element; in particular, the information verbally received on the errors committed in a key element of the gymnastic performance leads to more gains in learning than others. Effects of learning new complex gymnastics routines depend on the content of feedback on task performance. Providing too much verbal feedback when learning gymnastic skills is less effective than limited verbal feedback on the errors of performing key elements.

Effects on learning new complex gymnastics routines depend on the type of feedback, amount and content of information, and complexity of the task. When learning a complex task using the progressive part method, it is recommended to deliver short guidelines on errors. Further research is needed to determine the principles of learning complex gymnastics routines.

Therefore, it is advisable to carry out studies into complex gymnastic elements and routines. When teaching gymnastic exercises to elite athletes, we should employ verbal feedback on errors in key elements, whereas in the case of less advanced gymnasts, positive verbal feedback combined with visual feedback seems to be more beneficial. Novice gymnasts should be shown how to perform tasks accurately and then be provided with positive verbal feedback, while elite gymnasts can be given verbal feedback on errors only.

Considering the role of verbal feedback in teaching gymnastic exercises to elite athletes, the results of this study suggest employing verbal feedback on errors in key elements. In the case of less advanced gymnasts, positive verbal feedback combined with visual feedback seems to be more beneficial. Furthermore, novice gymnasts should be shown how to perform tasks accurately and then be provided with positive verbal feedback, while elite gymnasts can be given verbal feedback on errors only.

**Author Contributions:** Conceptualization, M.S. and T.N.; methodology, M.S., M.B., M.N. and T.N.; formal analysis, M.S., Ł.R. and J.J.-N.; resources, M.S., M.B. and M.N.; data curation, M.S.; writing—original draft preparation, M.S., M.B., M.N., T.N., T.A., Ł.R. and J.J.-N.; writing—review and editing, M.S., M.B., M.N., T.N., T.A., Ł.R. and J.J.-N. All authors have read and agreed to the published version of the manuscript.

**Funding:** This research received no external funding.

**Institutional Review Board Statement:** Not applicable.

**Informed Consent Statement:** Not applicable.

**Data Availability Statement:** All data are presented in the study.

**Conflicts of Interest:** The authors declare no conflict of interest.

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
