# Peer review of "The Role of Verbal Feedback in the Motor Learning of Gymnastic Skills: A Systematic Review"

_applsci, doi:10.3390/app12125940_

Round 1
Reviewer 1 Report
Dear Authors
We read with great interest the manuscript entitled “ The role of the verbal feedback in motor learning of the gym- 2 nastic skills: a systematic review”
Please find hereafter the corrections needed prior to manuscript resubmission. Some required revisions are absolutely mandatory before resubmission.
Materials and Methods
Search strategy. Please assess the pecisely period of time that searched databases were collected.
Best whishes

Author Response
Dear Reviewer,
Thank you very much for your time and valuable comments, which all have been considered and incorporated. The detailed list of responses is given below. We hope that the modifications and explanation will be acceptable for you.
Yours sincerely,
Rydzik, corresponding author
Dear Authors
We read with great interest the manuscript entitled “ The role of the verbal feedback in motor learning of the gym- 2 nastic skills: a systematic review”
Please find hereafter the corrections needed prior to manuscript resubmission. Some required revisions are absolutely mandatory before resubmission.
Materials and Methods
Search strategy. Please assess the pecisely period of time that searched databases were collected.
Best whishes
Response 1: According to reviewers’ suggestions, manuscript has been reviewed and edited by university native English speaker.
Response 2: Thank you for this suggestion. We have decided to adopt a without time restriction strategy. Typically, we do not justify the period of time of included studies, unless we have some good reasons to support this decision. For example, if we have a specific scientific question e.g. if the research question is about the efficiency of a given type of training on gymnastic element that appeared after 2000, you can include only the studies published after 2000. It is actually ideal to start with no time restriction.
Reviewer 2 Report
Thank you for submitting this study. While the conducted systematic review has some promise, it currently comes across as of low quality. Consider checking and following the PRISMA guidelines, re-structuring the manuscript to enhance the quality of reporting, and conducting further analysis. Please see the attached document for more feedback.

Author Response
Dear Reviewer,
Thank you very much for your time and valuable comments, which all have been considered and incorporated. The detailed list of responses is given below. We hope that the modifications and explanation will be acceptable for you.
Yours sincerely,
Rydzik, corresponding author
Overall:
- Please proofread this paper as there are grammar mistakes (e.g., line 14).
Response: According to reviewers’ suggestions, manuscript has been reviewed and edited by university native English speaker.
Abstract:
- Consider adding inclusion criteria to help the reader and report quant results of the systematic review. Was is amateurs? Elite performers? Youth? Adults? Since no results are shown, line 17 “Results also support…” does not make sense.
Response: According to reviewers’ suggestions, we revised the Abstract section.
Introduction
- Lines 24-25: it is not clear why the sport is becoming more challenging.
Response: The gymnastics literature and competition programs suggest that the elements and routine are more complex and difficult. The gymnasts must compete with exercise routines according to the Code of Points (CoP) rules set by the International Gymnastics Federation Rhythmic Gymnastics Technical Committee (FIG-RC-TC) every four years. As the CoP changes for every Olympic cycle, the routines become more challenging and the requirements for the choreography become increasingly difficult.
- Para 1: no references are provided to support these points. Transition and flow between the ideas is hard to follow.
Response: This has been changed.
- Line 77: What does KP stand for? The idea of Knowledge of Performance hasn’t been introduced yet.
Response: This has been changed.
Methodology:
- Line 82: please make the search strategy a separate section that is clear to identify. The same is true for other key sections (i.e., inclusion/exclusion criteria, etc.). All this key information is mixed together into one section.
Response: We revised and organized sections in a more clearly way.
- Lines 90-91: what was disagreement rate?
Response: Cohen’s kappa statistics were utilized to determine inter-rater variability for the extraction process. Kappa agreement variables were not calculated for a title and abstract screening processes. We have added this information to the present version of the manuscript.
- Figure 1/line 106: Why does the title include “WoS”?
Response: This has been removed and changed.
- Lines 101-104: what as the agreement between the reviewers? How did you deal with conflict?
Response: All discrepancies between the reviewers were resolved through discussion and consensus. Reasons for exclusion of identified articles were recorded in all cases. If necessary, other author was involved in decision making (MS). The agreement between reviewers assessing the quality of studies was verified using a Cohen’s kappa correlation for risk of bias. We have added this information to the present version of the manuscript.
- Overall: what protocols were followed? Did you use PRISMA guidelines? If not, then the quality and scrutiny of the systematic review should be questioned. In general, consider re-structuring this section to follow a more standardised approach and provide all key information needed. See earlier comments.
Response: This has been changed.
- Well done for using the quality appraisal scores. Consider integrating the final score into the summary table 2 and deleting table 1.
Response: Thank you for this suggestion. However, we have decided to keep Table 1 unchanged. We consider that it is necessary to clearly report the missing items of each trial.
Results:
- Lines 130-133: Repetition of earlier stated information (see lines 89-90). Earlier stated info should be deleted.
Response: This has been changed.
- Table 2: “study design” column is redundant because your selection criterion was “randomised controlled trials”. Also, all studies reported sex rather than gender. Please use accurate language to describe this. There also are inconsistencies in reporting. For example, the sample described under Niznikowski & Sadowski (2020) includes participants’ sex. The same info is then repeated in the next column. A different approach is taken for other studies.
Response: We have revised Table 2 and corrected all missing or incorrect data.
- Overall: please list which studies are being referred to. Alternatively, a numbering system can be used to do so. For example, line 142 states that “In 8 studies…”. Which studies are these? You did so in line 148. Clarity and consistency of reporting is needed.
Response: This has been changed.
- The current write-up assembles a scoping review as very brief analysis was conducted. Given the selected studies were RCTs, further analysis could be conducted. I find Rienhoff et al.’s (2016) paper a good example of hwo to write-up the systematic review and conduct further analysis (see here: https://doi.org/10.1007/s40279-015-0442-4 )
Discussion:
- All discussion resembles the results section as it is mostly summaries of the results being provided. A deeper and more thorough analysis is needed.
Response: Thank you for your suggestions. We agree with the Reviewer. In the present version, we revised shortly a discussion section (first and fourth paragraph, limitations). Considering the weakness of the methodological quality and diversity of extracted studies, we decided to present a more brief and narrative synthesis of the findings in this systematic review.
- Limitations: very few limitations were identified. Currently, no guidelines for the systematic review were followed, no thorough analysis was conducted, only the RCT’s (I assume in English?) were included.
Response: We have revised the limitations of the study to reflect this comment.
Conclusions:
- Lines 319-320: It is very hard to see were “strong evidence” come from.
Response: We removed this sentence.
Reference list:
- Which studies listed here were selected for the final sample?
Response: We have revised this section (added and renumbered references).
Reviewer 3 Report
Doing a systematic review on motor learning I think is a good idea
Methodologically, the study is fine. Follow the correct instructions
In the summary it should be clear the methodological procedure followed and the results
Author Response
Dear Reviewer,
Thank you very much for your time and valuable comments, which all have been considered and incorporated. The detailed list of responses is given below. We hope that the modifications and explanation will be acceptable for you.
Yours sincerely,
Rydzik, corresponding author
Doing a systematic review on motor learning I think is a good idea
Methodologically, the study is fine. Follow the correct instructions
In the summary it should be clear the methodological procedure followed and the results
Response: Thank you for your suggestions. We have revised Methods and Results sections.
Round 2
Reviewer 2 Report
Thank you for making the recommended suggestions and improving the (communicated) quality of your work.